# Effect of Different Watering Regimes in Summer Season on Water Intake, Feed Intake, and Milk Production of Marecha She-camel (*Camelus dromedarius*)

**DOI:** 10.3390/ani11051342

**Published:** 2021-05-08

**Authors:** Asim Faraz, Naeem Ullah Khan, Annamaria Passantino, Michela Pugliese, Ecevit Eyduran, Carlos Iglesias Pastrana, Amir Ismail, Nasir Ali Tauqir, Abdul Waheed, Muhammad Shahid Nabeel

**Affiliations:** 1Department of Livestock and Poultry Production, Bahauddin Zakariya University, Multan 60800, Pakistan; drasimfaraz@bzu.edu.pk (A.F.); drabdulwaheed@bzu.edu.pk (A.W.); 2Institute of Dairy Sciences, University of Agriculture, Faisalabad 38000, Pakistan; gulmeri007@gmail.com; 3Department of Veterinary Sciences, Via Umberto Palatucci–98168 Messina, University of Messina, 98168 Messina, Italy; passanna@unime.it; 4Department of Business Administration, Faculty of Economics and Administrative Sciences, Iğdır University, Iğdır 76000, Turkey; ecevit.eyduran@igdir.edu.tr; 5Department of Genetics, Faculty of Veterinary Sciences, University of Córdoba, 14014 Córdoba, Spain; ciglesiaspastrana@gmail.com; 6Institute of Food Science and Nutrition, Bahauddin Zakariya University, Multan 60800, Pakistan; amirismail@bzu.edu.pk; 7Department of Animal Science, University of Sargodha, Sargodha 40100, Pakistan; tauqir.nasir@uos.edu.pk; 8Camel Breeding and Research Station Rakh Mahni, Punjab 30030, Pakistan; shahidnabeel007@gmail.com

**Keywords:** camel, watering restriction, milk production, milk composition

## Abstract

**Simple Summary:**

Camel livestock is a significant sector of agriculture in Pakistan. Punjab Marecha is the camel breed more diffuse for their production characteristics. In fact, this camel breed is well adapted to the desert ecosystem, tolerating elevated temperatures and dehydration. This study aimed to evaluate the effect of water deprivation on milk production performance in Marecha camels during the summer. Twelve she-camels in the early lactation stage were included. After seven days, camels were divided into three groups: Group 1 (G1) having access to water once a day, Group 2 (G2) having access once every 4 days, and Group 3 (G3) having access once every 6 days. Milk production and daily dry matter were decreased in camels deprived of water for 6 days. The results of the current study suggest that the camel is a productive animal even when subjected to water deprivation; however, its feed intake and milk production are decreased.

**Abstract:**

Twelve lactating healthy Marecha she-camels in the early lactation stage during the summer at Camel Breeding and Research Station Rakh-Mahni (Pakistan) were included. All animals were fed with *Medicago sativa* and *Cicer arientinum ad libitum* and divided into three groups in relation to the access to water, after a period of seven days of adaptation to experimental conditions. Group 1 (G1) was considered as control having access to water once every day; Group 2 (G2) had access once every 4 days, while Group 3 (G3) had access once every 6 days. The duration of the study was 60 days with an adaptation period to experimental conditions of 7 days before the onset of the study. Dry matter intake (DMI) was calculated on a dry matter basis. On average the ambient temperature and relative humidity during the trial were 39–41 °C and 55–63%, respectively. The DMI, water intake, milk production, and body weight changes were affected (*p* < 0.001) during various watering regimes. The mean values of water intake were found to be 82.94 ± 1.34 L higher in G3 than G1 and G2.

## 1. Introduction

The camel can thrive in desert ecosystems very efficiently compared with other livestock species [1,2]. They are well adapted and tolerant to thermal stresses due to their unique physiology and adaptation [3,4]. The camel can fluctuate its body temperature and conserve body water by minimizing evaporative loss by elevating body temperature up to 7 °C in the daytime [5]. This elevation mitigates the need for heat loss by panting/sweating, while the extra heat is dissipated during the night, which lowers the body temperature [6].

Various factors, including water, may influence milk production [7], but differently to other species such as ruminants, the water restriction does not seems to determine the quality of milk in camels [8,9]. Camels can survive with 30% dehydration and regain this loss by taking huge quantities of water, up to 100 L, in a limited time, having a more advanced water balance system than other desert livestock species [10]. Indeed, the camel can produce milk through their peculiar characteristics that ensure the maintenance of milk production, also decreasing food intake and increasing milk osmolality in crucial circumstances [11]. In deserts, dehydrated camels can provide adequate nutrition due to the water and salt contents of milk, which impart high quality for pastoral people [12].

Faced with water shortage, camels can maintain their metabolism, and their feed digestibility is not affected. The specialized metabolic and physiologic adaptation allows the camel to survive water deprivation periods without perishing [13].

As water provision is very imperative in deserts, a trial was conducted to assess the effect of water deprivation on milk production and water and feed intake in Marecha she-camels living in the Thal desert.

## 2. Materials and Methods

### 2.1. Study Area

The study was conducted during the summer season at Camel Breeding and Research Station (latitude 27.97010; longitude 73.36086) (CBRS), sited in Thal desert where the climate is continental–subtropical with temperature ranging between 45.6 and 5.5 °C (respectively, in summer and winter), while mean annual rainfall ranges between 151 and 355 mm [14].

### 2.2. Study Protocol

Twelve lactating healthy Marecha she-camels in the early lactation stage (first quarter) were divided into three groups (body weight average 586 ± 7.54 kg, milk production average 4.39 ± 0.09 kg). Each group was constituted by subjects and fed with alfalfa (*Medicago sativa*) and chickpea (*Cicer arientinum) ad libitum* (Table 1) [15].

The duration of the study was 60 days with 7 days as an adaptation period. Various watering regimes were followed: group G1 (Control) having access to water access daily for 30 min, G2 having access to water once every 4 days for 30 min, and G3 having access to water once every 6 days for 30 min.

During each phase of the study, milk production was recorded daily, and milk composition was analyzed at the end of each phase of water deprivation, and results were compared with the control (G1).

### 2.3. Camel Selection and Adaptation

Animals’ physiological parameters (respiration and pulse rate, rectal temperature), feed and water intake, and body weights were recorded in this acclimatization period. Health status was monitored, and all experimental animals were negative for endo/ectoparasite infections and trypanosome infection.

### 2.4. Daily Routine Practices

Morning milking was done daily by hand technique at 7:00 a.m. Physiological parameters were recorded immediately after milking. Calves were allowed to suckle cows for one minute and then separated. After milking, lactating she-camels wore an udder net to prevent calf suckling. Feeding ration to lactating camels was offered at 10:00 am. Watering to control group (G1) animals was done on a daily basis, while G2 and G3 were given water according to their protocol. Evening hand milking was performed at 4:00–5:00 p.m. Animals were allowed to move freely in a covered paddock (length 295 ft, width 203 ft) with an open area (length 136 ft, width 50 ft) from 5:00 to 9:00 p.m. Later on, they were housed in their respective pens (length 24.50 ft, width 12.75 ft).

### 2.5. Milk Composition, Feed and Water Intake

At the end of the water deprivation period, the milk samples were collected and immediately transferred to Milk Laboratory CBRS Rakh Mahni. The milk analysis was performed using standard protocols with an apposite instrument (Milky-Lab, Astori, Italy). Feed intake (DMI) was calculated on a daily basis, and dry matter percentage was determined by standard methods [16]. Water intake was measured by using a graduated tank. The air temperature and relative humidity were recorded twice daily (6:00 a.m., 12:00 p.m.) by hygrometer (Sling Psychrometer, Bacharach, Inc., USA; range of ambient temperature measured: −5 to 50 °C; range of humidity measured: between 10 and 100%).

### 2.6. Physiological Parameters, Body Weight, and Water Intake

Rectal temperature was noted twice daily by digital thermometer in the morning at 7:30 a.m. and in the evening at 5:00 p.m. just after milking. At the same times, the respiratory rate was monitored by flank movements, while pulse rate was recorded from coccygeal artery. The mean values of respiratory and pulse rate were obtained by five repeated measurements. Body weight changes were calculated before the beginning of the study (T0) and after the 15th (T1), 30th (T2), 45th (T3), and 60th days (T4) by digital weighing scale (Impressum, Gujranwala, Pakistan).

### 2.7. Statistical Analysis

Data collected were analyzed using a statistical software package (Minitab Statistical Software, Minitab, Ltd., London, UK) [17]. Data were expressed as least square means (LSMs) and standard error (SEM). The comparisons between the groups were performed using the analysis of variance on repeated measures. *p* < 0.05 was considered significant.

## 3. Results

During the study, the average temperature and humidity recorded were 39–41 °C and 55–63%, respectively.

Watering regimes affected (*p* < 0.01) water intake in lactating she-camels of G1, G2, and G3, with intake being higher in G1 (Table 2).

Water contents from feed were also a source of water for experimental camels during water deprivation. Maximum water intake was found in G3 followed by G2 and G1. Intake of succulent fodder was higher in G3 during the water deprivation period as compared to gram straw (Table 2).

Watering regimes affected (*p* < 0.01) total dry matter intake (DMI). Maximum DMI (10.23 kg) was found in G1 having access to water once a day, followed by G2 (9.88 kg) and G3 (8.77 kg). Daily dry matter intake was decreased when camels were deprived of water for 6 days (Table 2).

Watering regimes had a nonsignificant effect on dry matter water intake ratio from the feed. Maximum dry mater water intake ratio was found in G3 (deprived of water for 6 days), followed by G2 and G1 (Table 2).

Milk production was decreased as water deprivation in camel was prolonged, and this effect was significant (*p* < 0.01) (Table 3).

Watering regimes had a significant (*p* < 0.01) effect on milk fat percentage. The values of milk fat (%) were found to be increased in G2 and G3 as compared to G1. A higher level of fat (%) was reported in camels facing deprivation of 6 days (Table 3).

Watering regimes affected (*p* < 0.01) milk protein percentage and mean values for G1, G2, and G3, being higher in animals facing deprivation of 6 days (Table 3).

Watering regimes had a nonsignificant effect on lactose percentage. Lactose was improved in camels that were deprived of water, but this effect was nonsignificant (Table 3).

Watering regimes affected (*p* < 0.01) the milk total solids in G1, G2, and G3, being higher in animals of G2 and G3 (Table 3).

Watering regimes had a significant (*p* = 0.03) effect on milk density in G1, G2, and G3, being higher in G3 animals (Table 3).

Watering regimes had also a significant effect on the quantity of fat (*p* < 0.01), protein (*p* = 0.006), and total solids (*p* = 0.043) (Table 4).

During the study, the live body weight of the experimental animals was recorded at the end of the water deprivation period according to their different watering regimes, and results were compared within treatments. Body weight was found to be affected (*p* < 0.01) by water deprivation periods. No significant change was observed in camels having water access daily, while a weight loss was recorded in animals deprived of water for 4 and 6 consecutive days (Table 5).

## 4. Discussion

During the wet and rainy seasons in desert areas, camels mainly rely on browsing species moisture, as water is not available for several weeks to the pastorals [18]. Succulent fodder usually increases the milk yield. The intake of water from feed depends not merely upon the moisture level of feed but also on the amount of feed consumed. If the amount of feed consumed is ample, then water from the feed source will also be more for animals consuming that feed. Water restriction in animals, either for experimental purposes or naturally occurring, decreases the feed intake, and in this case, availability of water from feed will be lower [19], but in the current, study the feed intake did not decrease, probably due to the availability of lush green fodder as the animals fulfilled their water need for maintenance of feed intake.

Camels deprived of water for six consecutive days (G3) drank about 6 times more than G1 and compensated body loss after rehydration in just a few minutes, supporting the fact that camels can consume enough water in one instance to compensate for water losses during dehydration. Camels can survive without water for up to weeks in hot desert-like conditions, but feed intake is decreased during prolonged water deprivation. Camels decrease feed intake and metabolic rate during water deprivation or dehydration [7]. Bekele et al. (2011) [7] documented similar findings, showing that water intake increased in camels during the water deprivation span. Jakiimola (1995) [20] also observed the same findings, showing that water intake significantly increases up to 2 times after 7 consecutive days of water deprivation. DMI and water intake had a positive correlation. Current research findings match with the study of Siebert and MacFarlane (1975) [21], who reported that after dehydration, camels drink enough water to fulfill their losses during the dehydration period.

As feed intake was found to decrease as the water deprivation phase was prolonged, the present findings are in agreement with Guerouali and Wardeh (1998) [12], who reported that feed intake was increased at 10% dehydration but decreased by 27 and 44% at 15 and 20% of water deprivation, respectively. Rai et al. (1995) [22] observed depressed feed intake at days 18–20 of water deprivation in winter. DMI and water intake have a positive correlation, and this finding is supported by the study of Rai et al. (1995) [23], who found that dry matter intake decreased when prolonging the water deprivation to 4 days in Indian pack camels in hot and humid conditions.

Mousa et al. (1983) [24] reported that the daily DMI of the camel was found to be 2.38 ± 0.4 kg/day when water was offered. It was decreased to 1.90 ± 0.3 kg/day when water was deprived by 50%, and daily DMI was 1.5 ± 0.3 kg/day when camels faced 5 days of water deprivation. Konar and Thomas (1970) [25] and Engelhardt et al. (2006) [26] also reported the same findings, noting that when camels faced water deprivation for 11 days, dry roughage intake was decreased to 9.6%.

Research findings of Muna and Ammar (2001) [27] are in line with the results of the present study, as they reported that water is an important nutrient; if water intake was restricted, then feed consumption was also decreased in goats fed sorghum diet. Abdelatif et al. (2010) [28] found that feed intake is negatively affected by water restriction. The results of Alamer and Al-hozab (2004) [29] also support the current findings, as they showed that during water deprivation, feed intake decreased at a higher level in summer when the environmental temperature is high and unfavorable, followed by spring. Water deprivation had a negative effect on dry matter intake, as it is decreased when goats were deprived of water. Water intake and dry matter intake have a positive correlation.

Water is considered the most important nutrient for lactation. Thokal et al. (2004) [30], while working on crossbreed dairy cows farmed under different conditions, observed that milk production is decreased by 16% during water restriction in cattle where no significant level of total solids was changed during water restriction. Water deprivation has a negative effect on milk production; when increasing the increment of water, milk production is increased in dairy cattle at a significant level. The results of the present study showed that camels maintained their lactation level even when water was withheld for 6 days. One of the remarkable features of the camel is its ability to continue producing the same milk water content during the period of water deprivation or drought-like conditions when no water is available [11].

Current study results are in line with the findings of Turki et al. (2008) [31], who found that camel milk fat percentage is affected significantly by water restriction, while Yagil and Etizon (1980) [10] observed different results, showing that fat percentage, total solid content, protein, and lactose decreased.

Data suggest that the water restriction necessary to cause a dilution of camel milk is influenced by different mechanisms such as the type of diet and the number of days of water restriction.

Fat percentage in the current study is in range of the earlier studies in Pakistan [32], where milk fat percentage of 3.47–3.68% was found. Faraz et al. (2018) [33] documented milk fat, protein, lactose, total solids percentages of 3.95–4.90%, 2.75–3.95%, 3.87–5.10%, and 12.32–14.35%, respectively, in the Marecha dromedary camel and 3.88–4.70%, 2.66–4.02%, 3.67–5.04%, and 12.22–14.65%, respectively, in the milk of the Barela dromedary camel in desert conditions [34]. Raziq et al. (2011) [35] reported milk fat to be 2.5–2.8% in the Kohi white dromedary camel, which is lower than that found in the current study.

Khaskheli et al. (2005) [36] studied the Pakistani camel and found variations in milk composition, where total solids content ranged from 7.76 to 12.13%, protein ranged from 1.80 to 3.20%, and lactose ranged from 2.91 to 4.12%. The variations of current results may be attributed to the breed effect or different experiment conditions. Bekele et al. (2011) [7] documented that camel does not dilute its milk during the dehydration phase due to the change in milk and plasma osmolality during the water restriction phase. Results were also supported by Dahlborn (1987) [37] and Konar and Thomas (1970) [25], who observed higher milk fat percentages during water deprivation periods.

Zeleke (2007) [38] reported that drinking water did not affect lactose contents, and protein, fat, and dry matter were found to be higher in the dry season compared with wet months. Season and availability of water affected the proportion of fat (%) in camel milk. During the hot dry summer season, water shortage is problematic in the desert, and camel milk fat is found to increase. The results of the present study showed that milk protein and lactose contents were found to be better in dehydrated camels even after 6 days of water deprivation, even though no supplementation or concentrate was offered to camels as allowance.

On other hand, Dahlborn (1987) [37] reported a different scenario, as the milk protein contents were reduced during water restriction in camels, while the results obtained in the present study may be due to the animals being offered lush green fodder from which they had attained their necessary moisture levels. Similarly, Zeleke (2007) [38] agreed that the milk yield was higher during wet months, which may be due to accessibility of feed and water, as availability of water directly affects the milk yield in camels. Konar and Thomas (1970) [25] found that the milk composition, especially the lactose, changes with water restriction.

Maltz and Shkolnik (1984) [39], working on Bedouin goats, found that milk yield decreased by 35% of their initial value while osmolality of milk fat and protein percentage values increased when water was withheld for 4 days. Mengistu et al. (2007) [40], while working on water deprivation in Somali goats, found that milk and plasma osmolality increased during dehydration, supporting the current findings. Milk fat and total solids were increased during the water deprivation phase, but different findings were obtained regarding protein levels, as those were decreased. Milk yield was decreased in goats during water deprivation. Alamer (2009) [41] also reported that milk yield was decreased in goats when 25% and 50% of water was restricted. Total solids of milk were found to increase in goats after a 25% water restriction. Hossaini-Hilali et al. (1994) [42] agreed that water deprivation for 48 h resulted in a decline in milk yield in goats by 28%. Higher fat, protein, lactose, and total solids contents are considered as a source of a nutritious diet for growing calves, even in hostile environments where water is not available and vegetation is not available in surplus for providing required nutrients. These facts are also supported by the results of the present study, as the calves are able to attain ample nutrition in deep hostile deserts even in periods of scarcity.

Camels recompense their loss in body mass by consuming a huge amount of water at the time of rehydration after a prolonged water deprivation phase [17]. Prolonged water deprivation of up to 7 days has a significant effect on live body weight [20], and this loss progresses up to 8–16 days as well [19]. Moreover, water restriction has a negative effect on body condition scores [43]. The current result of 6 and 9% body weight loss in G2 and G3 is supported by the findings of the aforementioned authors.

Alamer (2006) [41] reported that local Saudi goats lost an average of 20.6% of their initial body weight after three days of water deprivation. The findings of Rai et al. (1995) [22] are in agreement with the present study; they showed that when Indian camels were deprived of water for 10 days, camels lost about 21.8% body mass. In another study by Rai et al. (1995) [23], water deprivation in camels during winter also resulted in body mass loss. Masoumi et al. (2011) reported findings that are in agreement with the current study results; they showed that camels lost about 10% body weight after 6 days of water deprivation [44].

About 40% of body weight was lost in Bedouin goats when they were deprived of water, and feed and water intake was decreased when water deprivation was increased [45]. In the present study, camels of G3 deprived of water for 6 consecutive days consumed about 6 times more water than control; this supports the fact that the camel can drink enough water in one instance to compensate for dehydration.

## 5. Conclusions

The findings of the current study indicate that the camel can be a productive animal even when subjected to water deprivation; however, its feed intake and milk production are decreased. Watering after 4 days (G2) is the most suitable watering regime for milk production in the Marecha camel. The results suggest that under arid environmental conditions, the camel is capable of providing food security to pastoralists, although the results of the present study are limited to the desert ecology of CBRS.

## Figures and Tables

**Table 1 animals-11-01342-t001:** Chemical composition of feeding species.

Feed/Forage Species	*Cicer Arientinum*	*Medicago Sativa*
Dry Matter (DM)	94.02	18.6
Crude Protein (CP)	9.62	23.01
Ether Extract (EE)	2.82	1.81
Crude Fiber (CF)	45.3	25.4
Neutral Detergent Fiber (NDF)	69.1	43.2
Acid Detergent Fiber (ADF)	48.7	30.2
Crude Ash (CA)	8.2	11.9

**Table 2 animals-11-01342-t002:** Effect of different watering regimes on water intake, feed intake (DM basis), and daily dry matter intake (DDMI).

Parameters	G1		G2		G3		*p*
	LSD	SEM	LSD	SEM	LSD	SEM	
Water intake (L)	18.33	0.54	47.65	1.36	82.84	1.36	<0.01
Water intake (feed) (L)	8.62	1.12	10.66	1.59	12.4	1.22	<0.01
DDMI (kg)	10.23	0.33	9.88	0.44	8.77	0.36	<0.01
Water intake (L):DMI	1.78:1	-	4.83:1	-	9.45:1	-	Ns

G1, Group 1; G2, Group 2; G3, Group 3; LSD, least significant difference; SEM, standard error of mean; DDMI, daily dry matter intake; DMI, total dry matter intake; Ns, not significant.

**Table 3 animals-11-01342-t003:** Effect of different watering regimes on milk production and composition.

Parameters	G1		G2		G3		*p*
	LSD	SEM	LSD	SEM	LSD	SEM	
Milk Production (L)	4.50	0.25	4.00	0.20	3.80	0.75	<0.01
Fat (%)	3.53	0.04	3.86	0.03	4.12	0.04	<0.01
Protein (%)	3.40	0.03	3.55	0.03	3.61	0.35	<0.01
Lactose (%)	4.62	0.02	4.78	0.02	4.80	0.02	Ns
Total Solids (%)	10.39	0.02	11.56	0.01	12.68	0.07	<0.01
Density (gm/cm^3^)	1.24	0.005	1.26	0.007	1.28	0.005	0.03

G1, Group 1; G2, Group 2; G3, Group 3; LSD, least significant difference; SEM, standard error of mean; Ns, not significant.

**Table 4 animals-11-01342-t004:** Effect of different watering regimes on quantity of fat, protein, lactose, and total solids in the milk.

Parameters	G1		G2		G3		*p*
	LSD	SEM	LSD	SEM	LSD	SEM	
Fat	15.88	0.18	15.44	0.80	15.7	0.15	<0.01
Protein	15.83	0.09	14.2	0.12	13.7	1.33	0.006
Lactose	20.8	0.09	19.13	0.12	18.2	0.08	Ns
Total Solids	46.75	0.09	46.24	0.04	48.2	0.03	0.043

G1, Group 1; G2, Group 2; G3, Group 3; LSD, least significant difference; SEM, standard error of mean; Ns, not significant.

**Table 5 animals-11-01342-t005:** Effect of different watering regimes on body weight changes in camels.

Time	G1		G2		G3	
	LSD	SEM	LSD	SEM	LSD	SEM
T1	0.00	0.00	32.00	0.71	44.75	1.97
T2	0.00	0.00	30.50	1.32	45.50	1.32
T3	0.00	0.00	34.50	0.87	43.75	1.03
T4	0.00	0.00	31.00	0.41	46.00	1.29

G1, Group 1; G2, Group 2; G3, Group 3.

## Data Availability

All the relevant data is available in the paper.

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
