# Peer review of "Effect of Different Watering Regimes in Summer Season on Water Intake, Feed Intake, and Milk Production of Marecha She-camel (Camelus dromedarius)"

_animals, 2021, doi:10.3390/ani11051342_

Round 1

Reviewer 1 Report

The paper is interesting because former reference stated that camel could increase its milk production after dehydration (see Yagil and Etzion). The present experiment is proving that it cannot be the case. However some comments can be done:

  1. The introduction can be improved. Firstly some comments are out of subject (lines 46-51 and also 55-61). The paper is focused on the i:mpact of watering rythm on milk production and composition, not on the camel demography or adaptation to heat. The introduction must introduce this subject and recall the objectives which are not appearing.
  2. Regarding the body weight change, it is not clear because in the table 3, only weight at d1 to d4 are reported. What's mean: the animals were weight only 4 days while the experiment was 60 days? Please explain in M&M the rythm of weighing (every week? every 2 weeks?)
  3. Some of parameters (milk production, water intake) were measured regularly. The authors did not mention if they took in account in variance model. Such data require variance analysis on repeated measure.
  4. the water intake must be also given by daily average to allow comparison. In deed  Water intake was 18,33/day (G1), 11,91/day (G2) and 13,8 (G3). Water in food must be added to have a true comparison. The maximum water intake in that case is in G1, not in G3...
  5. The concentration of fat, protein and all components in milk from dehydrated camel is due to the decrease of water in milk. To have a clear comparison, it should be better to compare the QUANTITY of fat, protein... exported  in the milk (i.e. by multiplying fat/protein concentration by the milk production) to assess if dehydration affect only water in milk or also the other components (this point could change the statement that the milk composition is "better" in dehydrated camels as stated in lines 274-276).
  6. Lines 166-168 must be moved in material and methods.
  7. Figures 1 and 2 must be deleted as they are copying the tables
  8. Lines 214 and 222: repetition. Moreover, there was no calculation of the correlation between DMI and water intake
  9. Lines 246-248: the result of Yagil and Etzion is not in line of the present results as a loss of water in milk is observed in the present study. This point must be discussed as the results of Yagil and Etzion are quite debatable.

Author Response

Dear Reviewer,

Thank you very much for your time and all your comments. We have revised the references and increased them. The English structure and grammar of the manuscript has been thoroughly reviewed.

 We thank for precise and thoughtful comments and constructive criticism, which has led to a better manuscript.

We revised the manuscript in relation to the suggestions and more detailed answers are given below.

The changes made in the manuscript to address comments are reported in red.

We do hope that the revised manuscript now suits for publication in Animals, Special Issue Trends in Camel Health and Production

  1. The introduction can be improved. Firstly, some comments are out of subject (lines 46-51 and also 55-61). The paper is focused on the impact of watering rhythm on milk production and composition, not on the camel demography or adaptation to heat. The introduction must introduce this subject and recall the objectives which are not appearing.

The introduction has been improved focusing on the impact of watering rhythm on the production and composition of the milk.

  1. Regarding the body weight change, it is not clear because in the table 3, only weight at d1 to d4 are reported. What's mean: the animals were weight only 4 days while the experiment was 60 days? Please explain in M&M the rhythm of weighing (every week? every 2 weeks?).

The rhythm of weighing has been expressed in materials and methods.

  1. Some of parameters (milk production, water intake) were measured regularly. The authors did not mention if they took in account in variance model. Such data require variance analysis on repeated measure.

The information has been reported in the text.

  1. The water intake must be also given by daily average to allow comparison. In deed  Water intake was 18,33/day (G1), 11,91/day (G2) and 13,8 (G3). Water in food must be added to have a true comparison. The maximum water intake in that case is in G1, not in G3.

The sentence has been corrected.

  1. The concentration of fat, protein and all components in milk from dehydrated camel is due to the decrease of water in milk. To have a clear comparison, it should be better to compare the QUANTITY of fat, protein... exported  in the milk (i.e. by multiplying fat/protein concentration by the milk production) to assess if dehydration affect only water in milk or also the other components (this point could change the statement that the milk composition is "better" in dehydrated camels as stated in lines 274-276).

The comparison on the milk quantity has been performed.

  1. Lines 166-168 must be moved in material and methods.

Lines 166-168 have been moved in material and methods.

  1. Figures 1 and 2 must be deleted as they are copying the tables

Figures 1 and 2 have been deleted.

  1. Lines 214 and 222: repetition. Moreover, there was no calculation of the correlation between DMI and water intake.

The sentence has been deleted.

  1. Lines 246-248: the result of Yagil and Etzion is not in line of the present results as a loss of water in milk is observed in the present study. This point must be discussed as the results of Yagil and Etzion are quite debatable.

The point has been discussed.

Reviewer 2 Report

In my opinion paper is corectly structured with respect to the object analyzed. The paper has a corectly basis of literature. However, I find that the some concept are not always sufficiently presented. Materials and Methods should be supplemented with additional information and explanations. For  example measurements of physiological parameters were not adequately described both in the methodology and in the results and discussion. A main part results is repeated information includes both  in tables and figures. The description of the results duplicates the data provided in the table. Section should be reorganized and indicate the relationships between the factors analyzed and the test results obtained. Also, no results of physiological studies on camels are given. This should be supplemented. Some of the conclusions do not come from the research presented. This needs to be changed. Specific comments in attached file.

Author Response

Dear Reviewer,

Thank you very much for your time and all your comments. We have revised the references and increased them. The English structure and grammar of the manuscript has been thoroughly reviewed.

 We thank for precise and thoughtful comments and constructive criticism, which has led to a better manuscript.

We revised the manuscript in relation to the suggestions and more detailed answers are given below.

The changes made in the manuscript to address comments are reported in red.

We do hope that the revised manuscript now suits for publication in Animals, Special Issue Trends in Camel Health and Production

Line 2-4: The title has been changed.

Line 56: The term environmental has been changed with the term suggested (termal).

Line 70-71: The sentence has been moved at the beginning of paragraph.

Line 72-74: The aim of the paper has been corrected specifying that data reported changes recorded in Marecha camels.

Line 77-78: GPS coordinates have been added.

Line 82-83: The body weight average of she-camels before the begin of experimental study has been added.

Line 94: The information has been added.

Line 104: Dimensions of the paddock have been added.

Line 105: Dimensions of the pens have been added.

Line 112: The sentence has been modified.

Line 112: The times of temperature and humidity have been added.

Line 112: The information about the device have been added.

Line 112-113: Temperature and humidity values has been reported in the results.

Line 116-117: The information required has been added.

Line 119: The information required has been added.

Line 121: The information required has been added.

Line 122: Data obtained are reported in tables.  The section has been reorganized.

Line 130-131: The sentence has been deleted.

Line 182: The figure 1 has been removed.

Line 191: The figure 2 has been removed.

Line 192: The table has been modified.

Line 205: “G3” has been added.

Line 141: The breed and farm conditions are clarified.

Line 321: The sentence has been deleted.

Line 324: The sentence has been modified.

Reviewer 3 Report

Dear Authors,

it is nice prepared manuscript. All comments and suggestion are in the text.

Author Response

Dear Reviewer,

Thank you very much for your time and all your comments. We have revised the references and increased them. The English structure and grammar of the manuscript has been thoroughly reviewed.

 We thank for precise and thoughtful comments and constructive criticism, which has led to a better manuscript.

We revised the manuscript in relation to the suggestions and more detailed answers are given below.

The changes made in the manuscript to address comments are reported in red.

We do hope that the revised manuscript now suits for publication in Animals, Special Issue Trends in Camel Health and Production

Line 34-35: The repetition has been delated.

Line 88-92: Data has been reported in table 1.

Line 117-121: The paragraph has been separated.

Line 164-165: The repetition has been delated.

Line 182: The figure 1 has been removed.

Line 191: The figure 2 has been removed.

Round 2

Reviewer 1 Report

All papers regarding other livestock species never started by general considerations on the number of animals in the world and variety of breed if the paper is focused on physiological aspect. So, why many authors working on camel  start their paper on camels by such statement? The paper is on water management by camel bot on the demography. Please, delete the first paragraph (lines 46-54) and statrt by consideration on the proverbial resistance of camel to dehydration and cite "Yagil, 1985. The desert camel. Comparative physiological adaptation. Karger Publ. London" rather your own publication which is speaking on food security rather than resistance to dehydration. On that topic, you can also eventually cite: "Bengoumi and Faye, 2002. Adaptation du dromadaire à la déshydratation. Rev. Sécheresse, 13, 121-129"

Line 64: various factors

Line 133: at the same time(s),

Please verify text of reference list (many errors: see some below)

Line 396: Camelus dromedarius in italic (also L. 433, 444, 445 etc...)

Line 403: chameaux et dromadaires, animaux laitiers

Line 404: Francaise

Line 453: Quality

Line 417: Options

Line 424: Indian

Author Response

Dear Reviewer,

Thank you very much for your time and all your comments. We have revised the references and increased them. 

 We thank you for your precise and thoughtful comments and constructive criticism, which has led to a better manuscript.

We revised the manuscript in relation to the suggestions and more detailed answers are given below.

The changes made in the manuscript to address comments are reported in red.

We do hope that the revised manuscript now suits for publication in Animals, Special Issue Trends in Camel Health and Production

REVIEWER 1

  1. All papers regarding other livestock species never started by general considerations on the number of animals in the world and variety of breed if the paper is focused on physiological aspect. So, why many authors working on camel  start their paper on camels by such statement? The paper is on water management by camel bot on the demography. Please, delete the first paragraph (lines 46-54) and start by consideration on the proverbial resistance of camel to dehydration and cite "Yagil, 1985. The desert camel. Comparative physiological adaptation. Karger Publ. London" rather your own publication which is speaking on food security rather than resistance to dehydration. On that topic, you can also eventually cite: "Bengoumi and Faye, 2002. Adaptation du dromadaire à la déshydratation. Rev. Sécheresse, 13, 121-129".

The first paragraph (lines 46-54) has been deleted and the references suggested have been included.

  1. Line 64: various factors

The sentence has been corrected.

  1. Line 133: at the same time(s).

The sentence has been corrected.

  1. Line 396: Camelus dromedarius in italic (also L. 433, 444, 445 etc...).

The references have been corrected.

  1. Line 403: chameaux et dromadaires, animaux laitiers

The reference has been corrected.

  1. Line 404: Francaise

The word has been corrected.

  1. Line 453: Quality

The word has been corrected.

Line 417: Options

  1. The word has been corrected.

Line 424: Indian

The word has been corrected.

Reviewer 2 Report

Dear Authors,

Thank you for your feedback. I believe that your amendments have improved the quality of the manuscript. Unfortunately, I still have some remarks regarding introduction and numbering of tables and figures. Detailed comments in attached file.

Author Response

Dear Reviewer,

Thank you very much for your time and all your comments. We have revised the references and increased them. T

 We thank you for your precise and thoughtful comments and constructive criticism, which has led to a better manuscript.

We revised the manuscript in relation to the suggestions and more detailed answers are given below.

The changes made in the manuscript to address comments are reported in red.

We do hope that the revised manuscript now suits for publication in Animals, Special Issue Trends in Camel Health and Production

Line 24: The word has been corrected.

Line 55-56: In accordance with reviewer 1, the sentence has been deleted.

Line 73: The sentence has been modified.

Line 150: The word has been corrected.

Lines 217-225: The number of tables has been corrected, the figures have been deleted.